# Contrastive Semi-Supervised Domain Adaptation with von Mises-Fisher for Class Imbalance Mitigation

## Abstract

In contrast to Unsupervised Domain Adaptation (UDA) methods that rely solely on unlabeled data, Semi-supervised Domain Adaptation (SSDA) aims to enhance classification accuracy and generalization by incorporating a small amount of labeled samples from the target domain. However, a central challenge in SSDA is effectively addressing the distributional discrepancy between the source and target domains, particularly when labeled target data is scarce. While existing SSDA approaches have alleviated this issue to some extent, the persistent problem of class imbalance remains a critical obstacle. To address this challenge, we propose a novel Contrastive Semi-Supervised Domain Adaptation (cSSDA) algorithm based on the von Mises-Fisher (vMF) distribution. The core idea is to integrate the vMF distribution into the contrastive learning framework to refine the contrastive loss, enabling the construction of an infinite number of contrastive pairs. This approach helps the model better handle the class imbalance inherent in SSDA. Specifically, the vMF distribution excels at modeling directional data in high-dimensional spaces, enhancing the model's ability to capture similarities and differences between source and target domains during contrastive learning. Extensive experiments conducted on two widely-used benchmark datasets demonstrate that our method consistently outperforms existing SSDA approaches.

## 1 Introduction

Unsupervised Domain adaptation (UDA)Li et al. (2024b); Kollias et al. (2024) has become a critical technique in machine learning which address the challenge of transferring knowledge learned from a source domain to a target domain, particularly when labeled data in the target domain is limited. While UDA methods have made significant progress by relying solely on unlabeled target data, Semi-supervised Domain Adaptation (SSDA)Ganin et al. (2016); Kang et al. (2019); Long et al. (2016); Na et al. (2021) presents a more practical approach by incorporating a small amount of labeled target data. Even though the labeled data is limited, it provides valuable supervision that can substantially improve classification accuracy and enhance the model's ability to generalize across domains.

In Semi-Supervised Domain Adaptation (SSDA), the goal is to bridge the domain gap between the source and target domains by leveraging labeled data from the source domain and a small amount of labeled target samples. The main challenge is effectively addressing the distributional discrepancy between these two domains. This issue becomes more evident when labeled target data is scarce, leading to an insufficient representation of certain classes and a class imbalance problem that complicates the adaptation process.

Existing SSDA methods have made significant progress in addressing various challenges by leveraging techniques such as consistency regularization, pseudo-labeling, and domain-invariant feature learning. For example, IDMNE Li et al. (2024a) introduces a novel cross-domain feature alignment strategy that mixes data at both the sample level and the manifold level between labeled source and target samples. CLDA Singh (2021) aims to bridge the intra-domain gap between labeled and unlabeled target data and the inter-domain gap between the source and unlabeled target domains. Full-T Li et al. (2021a) presents the first finite-sample generalization bounds for both classification

and regression tasks under the SSDA setting. This provides a principled approach to learning invariant representations while minimizing domain-specific risks, resulting in a theoretically grounded bound minimization algorithm. However, despite these advances, many existing approaches still struggle with the fundamental issue of sample imbalance, particularly in the target domain. This imbalance, often caused by insufficient representation of minority classes, significantly limits model performance. Models tend to be biased toward majority classes, leading to poor generalization to minority classes. Therefore, addressing class imbalance is crucial for achieving robust and equitable domain adaptation.

To further enhance the adaptability of the proposed cSSDA framework, we incorporate the von Mises-Fisher (vMF) distribution Kobayashi (2021) into the contrastive learning process, enabling the generation of a virtually infinite number of class-aware contrastive pairs. This design significantly improves representation learning, especially when labeled target samples are extremely limited. Unlike traditional contrastive methods that may suffer from biased or imbalanced sampling, the vMF-based contrastive formulation better captures the semantic structure of the feature space. It promotes both intra-class compactness and inter-class separability in the feature representations of both source and target domains. Moreover, our approach directly addresses the persistent challenge of class imbalance in SSDA. By leveraging the intrinsic properties of the vMF distribution, we construct a more balanced and structure-aware contrastive objective. The vMF distribution excels in modeling directional data, which is essential for improving the representation of minority classes in high-dimensional spaces. This leads to more equitable feature learning, where minority classes are better represented, ultimately improving the model's generalization ability across diverse and imbalanced categories. In summary, the vMF distribution improves our approach by refining contrastive learning in SSDA, addressing class imbalance, and ensuring that the model learns robust and discriminative representations even with limited labeled data.

- We propose cSSDA, a novel contrastive semi-supervised domain adaptation framework that introduces the von Mises-Fisher distribution into contrastive learning to better model directional data in high-dimensional spaces.
- We design a vMF-based contrastive loss that enables the construction of infinite contrastive pairs, alleviating the class imbalance issue and enhancing feature discrimination in both source and target domains.
- We conduct extensive experiments on two widely used SSDA benchmarks, Office-Home and DomainNet, demonstrating that our method consistently outperforms state-of-the-art SSDA approaches in terms of both overall accuracy and minority class performance.

## 2 METHOD

### 2.1 MOTIVATION AND PRELIMINARIES

While semi-supervised domain adaptation (SSDA) has made significant progress, the critical issue of class imbalance in SSDA remains understudied. To bridge this gap, we propose Contrastive Semi-Supervised Domain Adaptation (cSSDA), a novel approach that generates an infinite number of contrastive pairs to mitigate sample imbalance in SSDA. Our method is motivated by the observation that deep features on the unit sphere encode rich semantic information, where their statistical properties can effectively model intra-class and inter-class variations. By leveraging the von Mises-Fisher (vMF) distribution, cSSDA enables the sampling of infinite contrastive pairs, facilitating robust source-to-target domain transfer without explicit per-class sampling. Prior works, such as ProCo Du et al. (2024), have applied vMF distributions to long-tailed visual recognition, yet none have addressed the real-world challenge of imbalance in SSDA. Our work fills this gap by introducing a principled framework for handling domain adaptation under class imbalance.

In the SSDA problem, we are provided with three distinct datasets: a labeled source domain dataset denoted as $\mathcal{D}_s = \left\{ (x_i^s, y_i^s) \right\}_{i=1}^{N_s}$, a labeled target domain dataset $\mathcal{D}_l = \left\{ (x_i^l, y_i^l) \right\}_{i=1}^{N_l}$, and an unlabeled target domain dataset $\mathcal{D}_u = \left\{ (x_i^u,) \right\}_{i=1}^{N_u}$, where $N_s$, $N_l$ and $N_u$ represent the number of samples in each respective dataset. The primary objective of SSDA is to leverage $\mathcal{D}_s$, $\mathcal{D}_l$, and $\mathcal{D}_u$ to train a robust domain adaptation model, which is subsequently evaluated on the target domain data to assess its generalization performance under domain shift. This formulation addresses the practi-

cal scenario where limited labeled data is available in the target domain, while leveraging abundant labeled data from the source domain and additional unlabeled data from the target domain to bridge the domain gap.

## 2.2 SUPERVISED CONTRASTIVE LEARNING

Supervised Contrastive Learning (SCL) Khosla et al. (2020) extends contrastive learning to supervised scenarios by explicitly utilizing label information to learn discriminative feature representations. Given a batch of $N$ labeled samples $\{(x_i, y_i)\}_{i=1}^N$, SCL optimizes feature embeddings such that samples from the same class ($y_i = y_j$) are pulled together in the representation space while those from different classes are pushed apart. This is achieved through the following contrastive objective:

$$\mathcal{L}_{\text{SCL}} = \mathbb{E}_i \left[ \mathbb{E}_{p \in P(i)} \left[ -\log \frac{\exp(z_i^\top z_p / \tau)}{\sum_{a \in A(i)} \exp(z_i^\top z_a / \tau)} \right] \right] \tag{1}$$

where $z_i = f(x_i)$ denotes the feature embedding produced by encoder network $f$, $P(i) = \{j \in \{1, ..., N\} \mid y_j = y_i, j \neq i\}$ represents positive samples sharing the same label, $A(i) = \{1, ..., N\} \backslash \{i\}$ contains all other samples in the batch and $\tau > 0$ is a temperature hyperparameter controlling class separation.

The denominator's summation over $A(i)$ creates a dynamic set of negative samples, enabling more effective representation learning compared to traditional triplet losses. This formulation demonstrates superior performance in classification tasks by: (1) leveraging label information to construct more informative positive/negative pairs, and (2) benefiting from the contrastive learning framework's stability and efficiency.

SCL objective in Eq. 1 inherently assumes two conditions for effective optimization: (1) sufficiently large batches to ensure diverse positive/negative pairs, and (2) approximately uniform label distributions within batches. However, these assumptions often break down in SSDA scenarios. First, the limited availability of labeled target samples ($\mathcal{D}_l$) restricts batch diversity, while the typical long-tail distribution in real-world datasets (both in source $\mathcal{D}_s$ and target domains) leads to imbalanced class representations. When applied to SSDA, this imbalance causes SCL to disproportionately prioritize head classes from the source domain, where samples are abundant while underrepresenting tail classes and target-domain-specific features. Consequently, the model develops biased representations that degrade performance on minority classes, particularly for target-domain samples where supervision is already scarce. This issue is further exacerbated by domain shift, as contrastive pairs drawn from imbalanced batches may reinforce spurious cross-domain correlations rather than meaningful semantic relationships.

To overcome this challenge, we propose leveraging the von Mises-Fisher (vMF) distribution Banerjee et al. (2005), a natural choice for modeling directional data on unit hyperspheres. While conventional normal distributions assume unconstrained Euclidean space, the vMF distribution explicitly accounts for the geometric constraints of unit-norm feature representations that are fundamental to contrastive learning frameworks. This property becomes particularly crucial in SSDA settings with long-tailed distributions, where standard approaches fail to adequately represent minority classes due to their sparse samples. The vMF distribution addresses this limitation by: (1) respecting the hyperspherical geometry of normalized feature embeddings, (2) enabling precise modeling of class-specific distributions regardless of sample quantity, and (3) maintaining stable parameter estimation even for classes with limited instances. By incorporating vMF-based modeling into our SSDA framework, we achieve more robust feature representation learning that properly handles both domain shift and class imbalance simultaneously.

## 2.3 HYPERSPHERICAL FEATURE MODELING WITH vMF DISTRIBUTION

Following the standard practice in contrastive learning, we constrain the learned features to reside on a unit hypersphere $\mathcal{S}^{d-1} \subset \mathbb{R}^d$. To properly model such directional data, we adopt the von Mises-Fisher (vMF) distribution Mardia et al. (2000), which serves as a natural analogue of the Gaussian distribution for spherical manifolds. The vMF distribution for a $d$-dimensional unit feature vector

$\mathbf{z} \in \mathcal{S}^{d-1}$ is characterized by:

$$p(\mathbf{z}|\boldsymbol{\mu}, \kappa) = C_d(\kappa) \exp\left(\kappa \boldsymbol{\mu}^\top \mathbf{z}\right) \tag{2}$$

where $\boldsymbol{\mu} \in \mathcal{S}^{d-1}$ is the mean direction, $\kappa \geq 0$ is the concentration parameter and $C_d(\kappa)$ is the normalization constant.

$$C_d(\kappa) = \frac{\kappa^{d/2-1}}{(2\pi)^{d/2} I_{d/2-1}(\kappa)} \tag{3}$$

with $I_r(\cdot)$ denoting the modified Bessel function of the first kind. The concentration parameter $\kappa$ governs the distribution's spread. The concentration parameter $\kappa$ governs the distribution's spread: larger values of $\kappa$ indicate stronger concentration around the mean direction $\boldsymbol{\mu}$, while $\kappa = 0$ corresponds to a uniform distribution on the hypersphere.

Building upon this probabilistic framework, we model the feature distribution using a mixture of vMF distributions. Let $X = \{\mathbf{z}_i\}_{i=1}^N \subset \mathcal{S}^{d-1}$ be a set of unit-norm feature vectors extracted from our encoder. For maximum likelihood estimation of a single vMF component's parameters $(\boldsymbol{\mu}, \kappa)$, we assume the observations are drawn i.i.d. from $p(\mathbf{z}|\boldsymbol{\mu}, \kappa)$ as defined in Eq. 2. The likelihood function for $X$ is then given by:

$$
\begin{aligned}
P(X \mid \boldsymbol{\mu}, \kappa) &= \prod_{i=1}^n p(\mathbf{z}_i \mid \boldsymbol{\mu}, \kappa) \\
&= \prod_{i=1}^n C_d(\kappa) \exp\left(\kappa \boldsymbol{\mu}^\top \mathbf{z}_i\right),
\end{aligned}
\tag{4}
$$

where $C_d(\kappa)$ is the normalization constant defined in Eq. 3.

Taking the Eq. 4, we can derive the log-likelihood:

$$\ln P(X \mid \boldsymbol{\mu}, \kappa) = n \ln C_d(\kappa) + \kappa \boldsymbol{\mu}^\top \boldsymbol{r}, \tag{5}$$

where $\boldsymbol{r} = \sum_i \boldsymbol{z}_i$. To derive the maximum likelihood estimates (MLEs) of $\boldsymbol{\mu}$ and $\kappa$, we maximize the log-likelihood in Eq. 5 under the constraints $\boldsymbol{\mu}^T \boldsymbol{\mu} = 1$ and $\kappa \geq 0$. By introducing a Lagrange multiplier $\lambda$ for the norm constraint, the Lagrangian of the optimization problem becomes:

$$\mathcal{L}(\boldsymbol{\mu}, \kappa, \lambda; X) = n \ln C_d(\kappa) + \kappa \boldsymbol{\mu}^\top \boldsymbol{r} + \lambda(1 - \boldsymbol{\mu}^\top \boldsymbol{\mu}), \tag{6}$$

To derive the maximum likelihood estimates, we take partial derivatives of the Lagrangian (Eq. 6) with respect to each parameter and set them to zero, yielding the following system of equations for $\hat{\boldsymbol{\mu}}$, $\hat{\lambda}$, and $\hat{\kappa}$:

$$\hat{\boldsymbol{\mu}} = \frac{\boldsymbol{r}}{|\boldsymbol{r}|} \tag{7}$$

$$\hat{\kappa} = A_d^{-1}\left(\frac{|\boldsymbol{r}|}{n}\right) \tag{8}$$

$$\frac{C_d'(\hat{\kappa})}{C_d(\hat{\kappa})} = -\frac{\boldsymbol{r}^\top \boldsymbol{r}}{n|\boldsymbol{r}|} \tag{9}$$

Substituting the Lagrangian formulation in Eq. 6 into the first-order optimality conditions specified in Eq. 7, we obtain:

$$\hat{\lambda} = \frac{\hat{\kappa}}{2}\|\boldsymbol{r}\| \tag{10}$$

$$\text{and} \quad \hat{\boldsymbol{\mu}} = \frac{\boldsymbol{r}}{\|\boldsymbol{r}\|} = \frac{\sum_{i=1}^n \boldsymbol{z}_i}{\|\sum_{i=1}^n \boldsymbol{z}_i\|}. \tag{11}$$

Substituting $\hat{\mu}$ from Eq. 10, we can obtain

$$\frac{C_d'(\hat{\kappa})}{C_d(\hat{\kappa})} = -\frac{\|\boldsymbol{r}\|}{n} = -\bar{\boldsymbol{r}} \tag{12}$$

## 2.4 ONLINE ESTIMATION OF CLASS MEANS

We compute class means through an online updating scheme that incrementally aggregates statistics from both historical and current batch data. This approach utilizes two distinct mean estimates: (1) the persistent mean carried over from the previous epoch for model estimation, and (2) a running mean initialized anew each epoch for online adaptation. The update rule is given by:

$$\bar{z}_j^{(t)} = \frac{n_j^{(t-1)} \bar{z}_j^{(t-1)} + m_j^{(t)} \bar{z}_j'^{(t)}}{n_j^{(t-1)} + m_j^{(t)}}, \tag{13}$$

where $\bar{z}_j^{(t)}$ denotes the online mean estimate for class $j$ at iteration $t$, $\bar{z}_j'^{(t)}$ represents the batch mean of class $j$ in the current mini-batch, and $n_j^{(t-1)}$ and $m_j^{(t)}$ are the cumulative sample count before iteration $t$ and current batch size, respectively.

While one could theoretically sample contrastive pairs directly from the estimated vMF mixture distribution, this approach presents significant computational inefficiencies during iterative training. Instead, we derive an exact analytical solution by considering the asymptotic case where the number of samples approaches infinity. Through rigorous mathematical analysis, we obtain a closed-form expression for the expected contrastive loss that completely avoids the need for stochastic sampling.

## 2.5 CONTRASTIVE ALIGNMENT

Building upon the supervised contrastive loss definition in Eq. 1, we decompose it into two terms::

$$\mathcal{L}_{\text{SCL}} = \mathbb{E}_i \left[ \mathbb{E}_{p \in P(i)} \left[ -\frac{z_i^\top z_p}{\tau} \right] \right] + \mathbb{E}_i \left[ \log \sum_{a \in A(i)} \exp \left( \frac{z_i^\top z_a}{\tau} \right) \right] \tag{14}$$

For the positive term, we take the expectation over the vMF distribution:

$$\mathbb{E}_{p \in P(i)} \left[ z_i^\top z_p \right] = z_i^\top \mathbb{E}_{z_p \sim \text{vMF}(\mu_{y_i}, \kappa_{y_i})} [z_p] \tag{15}$$

$$= z_i^\top A_p(\kappa_{y_i}) \mu_{y_i} \tag{16}$$

where:

$$A_p(\kappa) = \frac{I_{p/2}(\kappa)}{I_{p/2-1}(\kappa)} \tag{17}$$

is the mean normalization factor for vMF distributions, with $I_{p/2}$ being the modified Bessel function of the first kind.

For the negative term, we consider the expectation over all samples as $N \to \infty$:

$$\mathbb{E} \left[ \log \sum_{a \in A(i)} \exp \left( \frac{z_i^\top z_a}{\tau} \right) \right] \tag{18}$$

$$= \log \left( \sum_{j=1}^K \pi_j \mathbb{E}_{z_a \sim \text{vMF}(\mu_j, \kappa_j)} \left[ \exp \left( \frac{z_i^\top z_a}{\tau} \right) \right] \right) \tag{19}$$

Using the moment generating function of vMF distributions:

$$\mathbb{E}_{z_a} \left[ \exp \left( \frac{z_i^\top z_a}{\tau} \right) \right] = \frac{C_p(\tilde{\kappa}_j)}{C_p(\kappa_j)} \tag{20}$$

where $\tilde{\kappa}_j = |\kappa_j \mu_j + z_i/\tau|_2$ and $C_p(\kappa)$ is the vMF normalization constant.

Combining both terms, we obtain the closed-form expected loss:

$$\mathcal{L}_{\text{SCL}} = \mathbb{E}_i \left[ -\frac{z_i^\top A_p(\kappa_{y_i}) \mu_{y_i}}{\tau} + \log \left( \sum_{j=1}^K \pi_j \frac{C_p(\tilde{\kappa}_j)}{C_p(\kappa_j)} \right) \right] \tag{21}$$

This expected form connects directly to the vMF-based analysis in our theoretical framework and enables efficient computation of the contrastive loss without explicit sampling.

## 2.6 Overall Formulation

Our framework extends SSDA Yu & Lin (2023) through two key components: (1) source label adaptation (SLA) and (2) contrastive feature alignment. Given labeled source data $S = \{(\mathbf{x}_i^s, y_i^s)\}$, labeled target data $L = \{(\mathbf{x}_i^t, y_i^t)\}$, and unlabeled target data $U = \{\mathbf{x}_i^u\}$, we optimize:

$$\mathcal{L}_{\text{cSSDA}} = \underbrace{\mathcal{L}_{\text{SLA}}}_{\text{Source Label Adaptation}} + \underbrace{\lambda \mathcal{L}_{\text{SCL}}}_{\text{Contrastive Alignment}} \tag{22}$$

Following the baseline SLA Yu & Lin (2023) approach, we adapt noisy source labels through:

$$\tilde{y}_i^s = (1 - \alpha) y_i^s + \alpha P_{\mathbf{C}_f}(\mathbf{x}_i^s) \tag{23}$$

where $P_{\mathbf{C}_f}$ is the prototypical classifier and $\alpha$ controls adaptation strength. The SLA loss becomes:

$$\mathcal{L}_{\text{SLA}} = \frac{1}{|S|} \sum_{i=1}^{|S|} H(g(\mathbf{x}_i^s), \tilde{y}_i^s) \tag{24}$$

and $H$ is the cross-entropy loss.

## 3 Experiments

### 3.1 Experiment Datasets

In order to better prove the performance of our model (cSSDA), We evaluate our model on two benchmark datasets, Office-Home Venkateswara et al. (2017) and DomainNet Peng et al. (2019). Office-Home dataset Venkateswara et al. (2017) is widely used in domain adaptation research for image recognition tasks. It consists of four distinct domains: Art (A), Clipart (C), Product (P), and Real World (R), capturing varying visual distributions. The dataset includes a total of 15,500 images distributed across 65 classes, providing a challenging benchmark for evaluating cross-domain learning algorithms. DomainNet dataset Peng et al. (2019) consists of 345 classes distributed across six distinct domains. These domains include Clipart (C), featuring clipart images; Real (R), containing photos and real-world images; Sketch (S), with sketches of specific objects; Infograph (I), comprising infographic images of various objects; Painting (P), presenting artistic depictions of objects in the form of paintings; and Quickdraw (Q), which includes drawings contributed by players of the game "Quick Draw." Following prior works Yang et al. (2021b); Li et al. (2021b); Yan et al. (2022), we evaluate our approach on four domains: Clipart (C), Painting (P), Real (R), and Sketch (S), using a representative subset of 126 classes.

Following established protocols in Yang et al. (2021b); Li et al. (2021b); Yan et al. (2022), we maintain consistent sampling strategies for both training and validation sets across datasets. Each dataset undergoes comprehensive evaluation through both one-pass and three-pass experimental procedures.

### 3.2 Comparative Experiments

Table 1: In the 3-shot comparison experiments on the Office-Home dataset, the best-performing results are indicated in bold.

| Method | A→C | A→P | A→R | C→A | C→P | C→R | P→A | P→C | P→R | R→A | R→C | R→P | Avg |
|---|---|---|---|---|---|---|---|---|---|---|---|---|---|
| S+T | 54.0 | 73.1 | 74.2 | 57.6 | 72.3 | 68.3 | 63.5 | 53.8 | 73.1 | 67.8 | 55.7 | 80.8 | 66.2 |
| DANNGanin et al. (2016) | 54.7 | 68.3 | 73.8 | 55.1 | 67.5 | 67.1 | 56.6 | 51.8 | 69.2 | 65.2 | 57.3 | 75.5 | 63.5 |
| ENTGrandvalet & Bengio (2004) | 61.3 | 79.5 | 79.1 | 64.7 | 79.1 | 70.2 | 62.6 | 85.7 | 71.9 | 73.4 | 66.4 | 86.2 | 74.0 |
| APEKim & Kim (2020) | 63.9 | 81.1 | 80.2 | 66.6 | 79.9 | 76.8 | 67.1 | 65.2 | 82.0 | 74.0 | 70.4 | **87.7** | 75.7 |
| DECOTAYang et al. (2021b) | 64.0 | 81.8 | 80.5 | 68.0 | **83.2** | 79.0 | 69.9 | 68.0 | 82.1 | 74.0 | 70.4 | **87.7** | 75.7 |
| MMESaito et al. (2019) | 63.6 | 79.0 | 79.7 | 67.2 | 79.6 | 76.6 | 65.5 | 64.6 | 80.1 | 71.3 | 64.6 | 85.5 | 73.1 |
| MME SLAYu & Lin (2023) | 65.9 | 81.1 | 80.5 | 69.2 | 81.9 | 79.4 | 69.7 | 67.4 | 81.9 | **74.7** | 68.4 | 87.4 | 75.6 |
| CDACLi et al. (2021b) | 66.7 | 79.0 | **83.6** | 66.7 | 78.0 | 80.0 | 64.1 | 67.2 | **86.2** | 68.7 | 69.7 | 86.2 | 74.7 |
| CDAC SLAYu & Lin (2023) | 65.6 | 81.4 | 81.1 | 68.2 | 82.1 | 80.1 | 67.7 | 68.9 | 82.6 | 69.0 | 69.7 | 86.3 | 75.2 |
| cSSDA(Ours) | **67.9** | **83.6** | 82.2 | **69.6** | 83.0 | **81.2** | **70.8** | **70.9** | 83.3 | 73.8 | **70.5** | 87.2 | **77.0** |

**Comparative Experiments on Office-Home:** We evaluate our method on the Office-Home dataset under both 1-shot and 3-shot settings, with the results presented in Table 1 and Table 2, respectively.

Table 2: In the 1-shot comparison experiments on the Office-Home dataset, the highest-performing results are highlighted in bold.

| Method | A→C | A→P | A→R | C→A | C→P | C→R | P→A | P→C | P→R | R→A | R→C | R→P | Avg. |
|---|---|---|---|---|---|---|---|---|---|---|---|---|---|
| S+T | 50.9 | 69.8 | 73.8 | 56.3 | 68.1 | 70.0 | 57.2 | 48.3 | 74.4 | 66.2 | 52.1 | 78.6 | 63.8 |
| DANNGanin et al. (2016) | 52.3 | 67.9 | 73.9 | 54.1 | 66.8 | 69.2 | 55.7 | 51.9 | 68.4 | 64.5 | 53.1 | 74.8 | 62.7 |
| ENTGrandvalet & Bengio (2004) | 52.9 | 75.0 | 76.7 | 63.2 | 73.6 | 70.4 | 53.6 | 81.9 | 67.9 | 72.5 | 60.7 | 81.6 | 68.9 |
| APEKim & Kim (2020) | 53.9 | 76.1 | 75.2 | 63.6 | 69.8 | 72.3 | 58.3 | 78.6 | 72.5 | 71.3 | 56.0 | 79.4 | 64.8 |
| DECOTAYang et al. (2021b) | 42.1 | 68.5 | 72.6 | 60.3 | 70.4 | 71.3 | 48.8 | 76.9 | 71.2 | 70.7 | 60.0 | 79.4 | 64.8 |
| MMESaito et al. (2019) | 59.6 | 75.5 | 77.8 | 65.7 | 74.5 | 74.8 | 64.7 | 57.4 | 79.2 | 71.2 | 61.9 | 82.8 | 70.4 |
| MME SLAYu & Lin (2023) | 62.1 | 76.3 | 78.6 | 67.5 | 77.1 | 75.1 | 66.7 | 59.9 | 80.0 | 72.9 | 64.1 | 83.8 | 72.0 |
| CDACLi et al. (2021b) | 61.2 | 75.9 | 78.5 | 64.5 | 75.1 | 75.3 | 64.6 | 59.3 | 80.0 | 72.7 | 61.9 | 83.1 | 71.0 |
| CDAC SLAYu & Lin (2023) | 61.4 | 77.8 | 79.2 | 66.9 | 76.2 | 75.9 | 66.3 | 60.6 | 80.5 | 71.6 | 65.6 | **84.3** | 72.2 |
| cSSDA(Ours) | **63.1** | **78.4** | **80.2** | **68.0** | **77.0** | **77.1** | **67.3** | **62.9** | **81.2** | **73.4** | **66.2** | **84.3** | **73.2** |

3-Shot Results (Table 1): In the 3-shot scenario, our proposed method SSPG achieves the highest average accuracy of 77.0%, surpassing several competitive baselines. For instance, on the A → C task, SSPG attains an accuracy of 67.9%, outperforming both CDAC SLA (65.6%) and DECOTA (65.3%). On the A → P task, SSPG achieves 83.6%, outperforming MME SLA, CDAC SLA, and other strong baselines. These results demonstrate cSSDA's robustness and its ability to effectively utilize limited labeled samples in semi-supervised domain adaptation.

1-Shot Results (Table 2): In the more challenging 1-shot setting, cSSDA continues to deliver strong performance with an average accuracy of 73.2%. Notable examples include the A → C task, where cSSDA achieves 63.1%, outperforming CDAC SLA (61.4%) and CDAC (61.2%).

Across both the 1-shot and 3-shot settings, cSSDA consistently outperforms or matches state-of-the-art methods such as MME SLA and CDAC SLA, highlighting its effectiveness and generalizability in semi-supervised domain adaptation, especially under low-resource conditions.

Table 3: In the 3-shot comparison experiments on the DomainNet dataset, the best-performing results are highlighted in bold.

| Method | R→C | R→P | P→C | C→S | S→P | R→S | P→R | Avg. |
|---|---|---|---|---|---|---|---|---|
| S+T | 60.0 | 62.2 | 59.4 | 55.0 | 59.5 | 50.1 | 73.9 | 60.0 |
| DANNGanin et al. (2016) | 59.8 | 62.8 | 59.6 | 55.4 | 59.9 | 54.9 | 72.2 | 60.7 |
| ENTGrandvalet & Bengio (2004) | 71.0 | 69.2 | 71.1 | 60.0 | 62.1 | 61.1 | 78.6 | 67.6 |
| APEKim & Kim (2020) | 76.6 | 72.1 | 76.7 | 63.1 | 66.1 | 67.8 | 79.4 | 71.7 |
| DECOTAYang et al. (2021b) | 80.4 | 75.2 | 78.7 | 68.6 | 72.7 | 71.9 | 81.5 | 75.6 |
| MMESaito et al. (2019) | 72.2 | 69.7 | 71.7 | 61.8 | 66.8 | 61.9 | 78.5 | 68.9 |
| MME SLAYu & Lin (2023) | 73.3 | 70.1 | 72.7 | 63.4 | 67.3 | 63.9 | 79.6 | 70.0 |
| CDACLi et al. (2021b) | 79.6 | 75.1 | 79.3 | 69.9 | 73.4 | 72.5 | 81.9 | 76.0 |
| CDAC SLAYu & Lin (2023) | 80.9 | 75.2 | 80.2 | 70.8 | 72.4 | **73.5** | 82.5 | 76.5 |
| cSSDA(Ours) | **82.0** | **76.4** | **81.2** | **72.1** | **74.1** | 73.8 | **82.9** | **77.5** |

Table 4: In the 1-shot comparison experiments on the DomainNet dataset, the best-performing results are marked in bold.

| Method | R→C | R→P | P→C | C→S | S→P | R→S | P→R | Avg. |
|---|---|---|---|---|---|---|---|---|
| S+T | 55.6 | 60.6 | 56.8 | 50.8 | 56.0 | 46.3 | 71.8 | 56.9 |
| DANNGanin et al. (2016) | 58.2 | 61.4 | 56.3 | 52.8 | 57.4 | 52.2 | 70.3 | 58.4 |
| ENTGrandvalet & Bengio (2004) | 65.2 | 65.9 | 65.4 | 54.6 | 59.7 | 52.1 | 75.0 | 62.6 |
| APEKim & Kim (2020) | 70.4 | 70.8 | 72.9 | 56.7 | 64.5 | 63.0 | 76.6 | 67.6 |
| DECOTAYang et al. (2021b) | 79.1 | 74.9 | 76.9 | 65.1 | **72.0** | 69.7 | 79.6 | 73.9 |
| MMESaito et al. (2019) | 70.0 | 67.7 | 69.0 | 56.3 | 64.8 | 61.0 | 76.1 | 66.4 |
| MME SLAYu & Lin (2023) | 71.8 | 68.2 | 70.4 | 59.3 | 64.9 | 61.8 | 77.2 | 68.8 |
| CDACLi et al. (2021b) | 77.4 | 74.2 | 75.5 | 67.6 | 71.0 | 69.2 | 80.4 | 73.6 |
| CDAC SLAYu & Lin (2023) | 79.2 | 75.2 | **77.2** | 68.1 | 71.7 | 71.7 | 80.4 | 74.8 |
| cSSDA(Ours) | **80.5** | **76.1** | **77.8** | **68.9** | 71.9 | **72.0** | **81.1** | **75.5** |

**Comparative Experiments on DomainNet:** We evaluate our method on the DomainNet dataset under both 1-shot and 3-shot settings, with results summarized in Table 3 and Table 4, respectively.

3-Shot Results (Table 3): In the 3-shot setting, our method cSSDA achieves the highest average accuracy of 77.5%, outperforming several strong baselines. Notable results include the R → C task, where cSSDA reaches 82.0%, surpassing CDAC SLA (80.9%) and DECOTA (80.4%). On the C → S task, cSSDA achieves 72.1%, demonstrating strong adaptability across challenging domain shifts. Overall, cSSDA consistently outperforms leading approaches such as MME SLA and CDAC SLA, underscoring its robustness and generalization capability.

1-Shot Results (Table 4): In the more challenging 1-shot setting, cSSDA continues to exhibit strong performance, achieving an average accuracy of 75.5%. For instance, on the R → C and R → P tasks, cSSDA obtains 80.5% and 76.1%, respectively, outperforming baselines such as DECOTA and CDAC SLA. These results further demonstrate cSSDA's effectiveness in extremely low-resource scenarios.

Across both 1-shot and 3-shot settings, cSSDA consistently delivers superior or comparable performance to state-of-the-art baselines. Its strong results under limited supervision highlight its potential for real-world semi-supervised domain adaptation applications, where labeled target data is scarce.

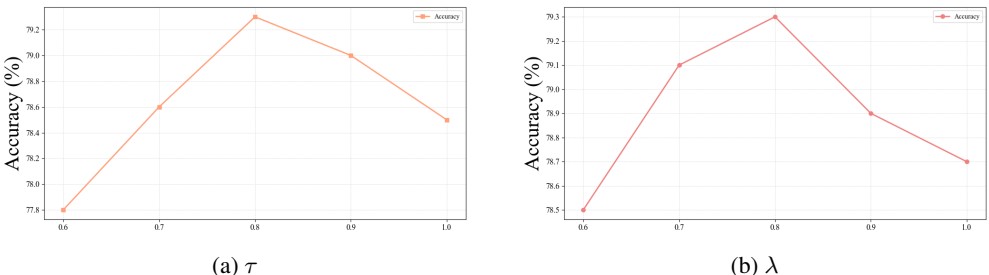

(a) $\tau$                      (b) $\lambda$

Figure 1: Accuracy and Loss comparison results shown in (a) and (b).

## 3.3 PARAMETER ROBUSTNESS ANALYSIS

To evaluate the robustness of our model with respect to parameter sensitivity, we conducted a series of parametric experiments using the Office-Home dataset, specifically focusing on 3-shot tasks. These experiments aimed to assess how variations in key parameters influence model performance, thereby identifying the optimal parameter ranges. We examined two critical parameters: the hyperparameter $\lambda$ (see Table 1(a)) and the temperature parameter $\tau$ (see Table 1(b)).

In each experiment, only one parameter was varied while all other parameters were maintained at their previously determined optimal values. This controlled approach ensured that the specific effect of each parameter on the model's performance could be accurately isolated and assessed without interference from other factors.

Table 5: Ablation study of our method

| Method | Source Proco | Target Proco | Accuracy (%) |
|---|---|---|---|
| Baseline | ✗ | ✗ | 65.6 |
| Method 1 (+source loss) | ✓ | ✗ | 67.24 |
| Method 2 (+target loss) | ✗ | ✓ | 67.36 |
| OURS (+source and +target) | ✓ | ✓ | 67.94 |

## 3.4 ABLATION STUDY

To rigorously evaluate the effectiveness of our proposed model, we conducted an ablation study using the Office-Home dataset under the 3-shot setting, as detailed in Table 5. This study aims to isolate and quantify the contributions of individual components within our framework. To substantiate the validity of our enhancements, we incorporated two previously established loss functions including ISDA Loss and Proco Loss into the ablation experiments. This inclusion directly compares with our newly proposed loss function, cSSDA Loss.

The results unequivocally demonstrate that our model consistently outperforms the baseline configurations and prior methodologies. Specifically, the integration of cSSDA Loss yields notable performance improvements, underscoring its efficacy in enhancing domain alignment and classification accuracy. As presented in Table 5, our model achieves superior accuracy across multiple tasks, thereby confirming that each introduced component positively contributes to the overall performance.

Moreover, the enhancements we propose exhibit significant transferability, enabling performance augmentation in other models. This characteristic not only reinforces the effectiveness of our approach within the current framework but also highlights its potential applicability to a broader spectrum of semi-supervised domain adaptation tasks.

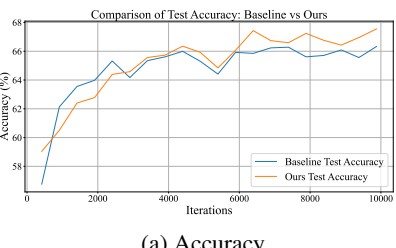 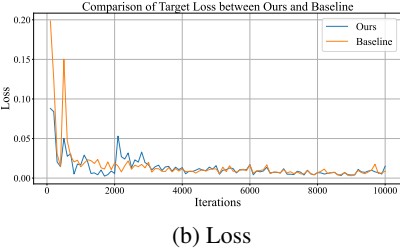

(a) Accuracy  (b) Loss

Figure 2: Accuracy and Loss comparison results shown in (a) and (b).

### 3.5 Loss Convergence and Test Accuracy Comparison

**Loss Convergence:** The loss convergence results are illustrated in Figure 2(a). In this figure, proco loss refers to our proposed loss function, denoted as $\mathcal{L}_{\text{SLA}}$. Our model demonstrates rapid convergence during training, reaching a stable state faster than the baseline models. This fast convergence indicates the efficiency of our optimization strategy, allowing the model to minimize both classification loss and domain alignment loss effectively within fewer iterations. The smooth downward trend in the loss curve further highlights the stability of our training process.

**Test Accuracy Comparison:** The test accuracy variation, presented in Figure 2(b), shows a comparison of our model's performance against CDAC SLA. As the figure demonstrates, our model consistently achieves superior accuracy throughout the training process, maintaining a clear margin over the baseline models. This consistency underscores the robustness of our approach, especially in handling domain shifts and generalizing to target samples. The steady accuracy curve indicates that our model is less prone to overfitting and performs reliably across various training epochs. These results confirm that our method not only converges efficiently but also delivers strong test performance in domain adaptation tasks.

## 4 Conclusion

In this work, we proposed a novel contrastive learning framework for semi-supervised domain adaptation (SSDA) that effectively addresses two critical challenges: domain shift and class imbalance. Our key innovation lies in leveraging the von Mises-Fisher (vMF) distribution to model feature representations on the unit hypersphere, enabling more robust feature learning compared to conventional Euclidean-space approaches. Specifically, the vMF-based formulation provides three main advantages: (1) geometrically-consistent modeling of normalized deep features, (2) stable estimation of class distributions regardless of sample quantity, and (3) natural generation of contrastive pairs that account for both inter-class and inter-domain relationships. Experimental results on benchmark datasets demonstrate that our method significantly outperforms existing SSDA approaches. Future directions include extending the vMF framework to multi-modal domain adaptation and investigating adaptive temperature parameters for the vMF distributions. The principles developed here may also benefit other imbalance-aware representation learning tasks beyond domain adaptation.

## 5 ETHICS STATEMENT

This work complies with the ICLR Code of Ethics. We present cSSDA, a framework for semi-supervised domain adaptation, evaluated on publicly available benchmark datasets. These datasets contain no personally identifiable or sensitive information, ensuring no risks to privacy or security. Our research advances energy-efficient semi-supervised domain adaptation with potential benefits for scientific and technological applications. All experimental protocols are transparently documented, with fair comparisons to prior work. The contributions are intended solely for research, supporting AI development.

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

APPENDIX

# A RELATED WORK

## A.1 DOMAIN ADAPTATION

Domain adaptation (DA) aims to bridge the distribution gap between a labeled source domain and an unlabeled (or sparsely labeled) target domain Huang et al. (2023); Xu et al. (2019); Zhuang et al. (2022); Zhang et al. (2022). The core challenge lies in learning domain-invariant representations while preserving discriminative semantic structures. Traditional DA approaches typically fall into two categories: feature distribution matching and semantic-conditional alignment. Early works minimized domain discrepancies through statistical measures like Maximum Mean Discrepancy (MMD) Pan et al. (2010); Yan et al. (2017) or moment matching Sun & Saenko (2016). For instance, Tzeng *et al.* Tzeng et al. (2014) aligned domain-specific features using single-kernel MMD, while Long *et al.* Long et al. (2015) extended this with multi-kernel MMD across multiple network layers. Subsequent adversarial methods Ganin et al. (2016); Tzeng et al. (2017); Su et al. (2020) further advanced this paradigm by employing domain discriminators to induce feature space alignment, as demonstrated by Saito *et al.*'s prototype-based clustering Saito et al. (2019). Recent studies Chen et al. (2019); Pan et al. (2019) reveal that global domain alignment may distort class-specific structures. This motivates semantic-aware adaptation that preserves categorical relationships. Works like Motiian et al. (2017); Li et al. (2021c) explicitly incorporate label information to guide finergrained feature alignment, showing superior transferability. Our approach builds on this insight but addresses a critical limitation: most methods assume clean source labels, while real-world adaptation requires handling label distribution shifts. Unlike prior works that either perform global domain alignment or assume perfectly aligned label spaces. We propose to enforce class-consistent feature learning across domains through our vMF-based contrastive loss. This jointly addresses domain shift and label shift while maintaining semantic discriminability.

## A.2 SEMI-SUPERVISED DOMAIN ADAPTATION

Semi-supervised domain adaptation (SSDA) achieves superior target-domain classification performance compared to unsupervised approaches by leveraging limited target labels Kim & Kim (2020); Li & Hospedales (2020); Chen et al. (2020); Yang et al. (2021a); Fang et al. (2023). While recent SSDA methods Saito et al. (2019); Kim & Kim (2020); Li et al. (2020); Qin et al. (2021); Li et al. (2021b) predominantly employ adversarial training for cross-domain alignment, several nonadversarial approaches have emerged: (1) Mishra et al. (2021) demonstrated that self-supervised pre-training with consistency regularization can enhance target classifiers without explicit domain alignment; (2) Luo et al. (2021) proposed Relaxed cGAN for image style transfer between domains; and (3) Fang et al. (2023) improved adaptation through intermediate style transfer between labeled and unlabeled samples. Beyond domain gap bridging, Yang et al. (2021a) innovatively decomposed SSDA into two complementary components: (1) semi-supervised learning (SSL) for target-domain discrimination enhancement, and (2) unsupervised domain adaptation for alignment optimization. Their framework employs dual classifiers trained with Mixup and Co-training respectively, learning mutually complementary features to boost adaptation performance. This approach parallels Zhong et al. (2021)'s contradictory learning strategy, where one classifier clusters target features to strengthen intra-class compactness and inter-class separation, while the other acts as a regularizer by dispersing source features to smooth decision boundaries.

# B COMPARISON METHODS AND SETTINGS

We conduct a comparative analysis against several representative baseline methods, including DANN Ganin et al. (2016), ENT Grandvalet & Bengio (2004), APE Kim & Kim (2020), DE-COTA Yang et al. (2021b), MME Saito et al. (2019), MME SLA Yu & Lin (2023), CDAC Li et al. (2021b), and CDAC SLA Yu & Lin (2023). Among these, CDAC SLA is adopted as the primary baseline for semi-supervised domain adaptation (SSDA), which utilizes only the labeled data from the source and target domains during training. DANN, a widely used method for unsupervised domain adaptation, is adapted to the SSDA setting by incorporating labeled target samples. ENT,

originally proposed for semi-supervised learning, serves as a classic entropy minimization strategy that promotes confident predictions on unlabeled data by minimizing the prediction entropy.

Our framework is designed to be compatible with various state-of-the-art semi-supervised domain adaptation (SSDA) methods. To validate its effectiveness, we adopt CDAC SLA Yu & Lin (2023) as the baseline model, enabling a fair and direct comparison with other leading approaches. All experiments are conducted using ResNet34 He et al. (2016) as the backbone network, pre-trained on the ImageNet1K dataset Deng et al. (2009). To ensure reproducibility and comparability, we follow the experimental protocols established in prior studies Li et al. (2021b); Saito et al. (2019), including the use of identical model architectures, batch sizes, learning rate schedules, optimization strategies, weight decay configurations, and initialization schemes.

For hyperparameters, MME and CDAC are configured using the settings recommended in their original papers. In experiments involving SLA, we set the mixing ratio $\alpha = 0.3$ and the temperature parameter $T = 0.6$. The model parameters are updated at intervals of 500 iterations. For MME, the warmup parameter $W$ is set to 500 for experiments on the Office-Home dataset and 3000 for experiments on DomainNet. For CDAC, the warmup parameter $W$ is set to 2000 on Office-Home and 5000 on DomainNet. The batch size is set to 24 in all experiments. After the warmup period, the learning rate scheduler is reset to enable label adaptation loss updates at a higher learning rate, allowing the model to adapt more effectively to evolving label distributions. For our improvement, we set the hyperparameters as follows: $\tau = 0.7$, and $\lambda = 0.2$, and the temperature is set to $\gamma = 0.8$ on Office-Home, $\gamma = 0.6$ on DomainNet.

All hyperparameters are carefully fine-tuned using validation performance to achieve optimal results. To ensure the robustness and reliability of our findings, each sub-task is executed over three independent runs, and the average performance is reported. For all baseline methods, we adhere strictly to the original implementation settings as described in their respective publications, thereby maintaining consistency and ensuring fair comparisons across all experimental evaluations.

## B.1 CONFUSION MATRIX COMPARISONS ANALYSIS

To provide an intuitive validation of our model's effectiveness, we analyze confusion matrices from the Office-Home A $\rightarrow$ C (3-shot) experiment. As shown in Figure 3, our model demonstrates significantly improved classification accuracy compared to the CDAC SLA baseline. The diagonal dominance in our model's confusion matrix highlights its enhanced discriminative capability, while the reduced off-diagonal misclassifications indicate better cross-domain alignment. This comparative visualization further substantiates the superior adaptation performance of our approach.

For this analysis, we randomly selected 10 categories from the 65 available in the Office-Home dataset. The trained models were utilized to predict the labels of the test samples within these selected categories. The confusion matrices provide a detailed representation of the relationship between the true labels and the predicted labels, thereby offering insights into the classification accuracy for each category. The confusion matrix provides a systematic visualization of classification performance, where diagonal entries correspond to correctly predicted instances (true positives) and off-diagonal elements represent erroneous classifications between categories. This structure enables precise identification of both model strengths and specific confusion patterns.

As depicted in Figure 3, our model exhibits higher accuracy, as evidenced by the more pronounced diagonal dominance in the confusion matrix compared to the baseline CDAC SLA. This enhanced diagonal dominance signifies that our model not only improves overall classification accuracy but also minimizes the number of misclassifications across various categories. The superior alignment of feature distributions between the source and target domains, facilitated by our model, plays a pivotal role in achieving this improvement. These findings substantiate the robustness of our approach in domain adaptation tasks, demonstrating its capability to generalize more effectively to target samples. The reduction in misclassification rates across categories underscores the efficacy of our model in enhancing domain alignment and classification performance.

## B.2 VISUALIZATION ANALYSIS

To provide an intuitive validation of our model's effectiveness, we conducted t-SNE visualizations for the A $\rightarrow$ C 3-shot domain adaptation task on the Office-Home dataset. Figure 4 compares the

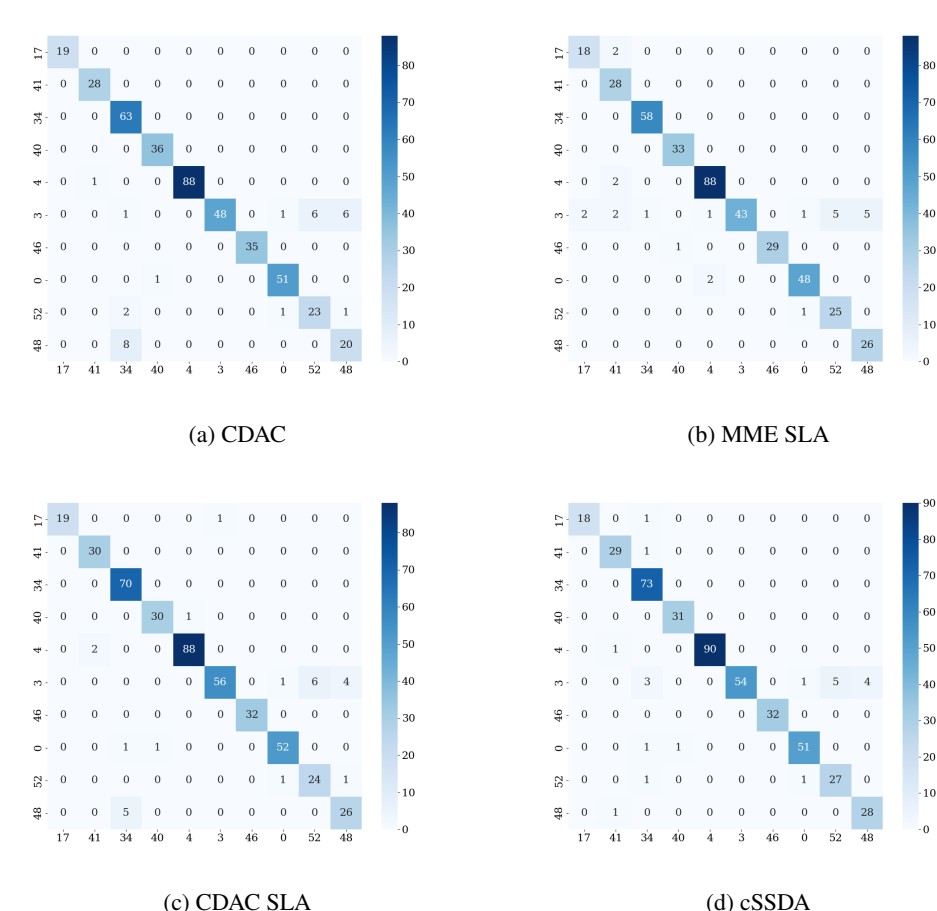

(a) CDAC

(b) MME SLA

(c) CDAC SLA

(d) cSSDA

Figure 3: Confusion Matrix Comparisons Analysis

feature space distributions generated by: (1) the baseline MME SLA Yu & Lin (2023), (2) standard CDAC Li et al. (2021b), (3) its self-training variant CDAC SLA Yu & Lin (2023), and (4) our proposed approach. The visualization clearly demonstrates our method's superior ability to learn discriminative and domain-invariant feature representations.

To visualize the feature distributions, we randomly sampled 10 out of 65 categories from the Office-Home dataset. Using our trained models, we extracted features and projected them into a 2D space via t-SNE. The visualization results demonstrate that our approach produces significantly more compact and distinct feature clusters compared to baseline methods. This improved clustering structure provides compelling evidence for our model's superior domain adaptation capability, as it reflects: (1) better intra-class compactness, indicating effective feature alignment across domains, and (2) enhanced inter-class separation, suggesting improved discriminative power. These characteristics directly contribute to the model's stronger generalization performance in the target domain.

## C  THE USE OF LARGE LANGUAGE MODELS (LLMs)

Large language models (LLMs) were only used to improve the clarity, grammar, and fluency of the manuscript. They were not involved in the development of research ideas, experimental design, data analysis, or any other aspect of the scientific content.

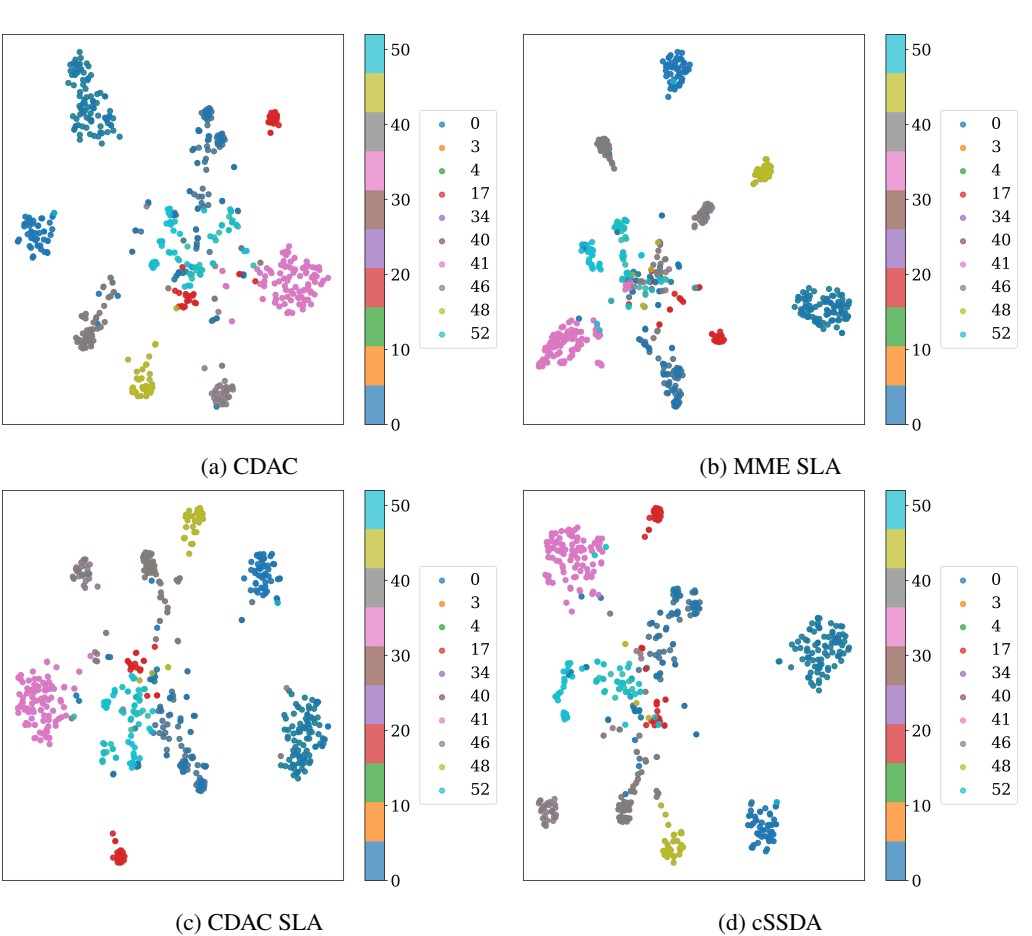

Figure 4: t-SNE dimensionality reduction visualization analyses