# OpenReview forum: "Contrastive Semi-Supervised Domain Adaptation with von Mises-Fisher for Class Imbalance Mitigation"
_ICLR.cc/2026/Conference — Submitted to ICLR 2026_

### Official Review · Reviewer_Ae4A · 2025-10-29

**Soundness:** 2
**Presentation:** 3
**Contribution:** 2
**Rating:** 2
**Confidence:** 5

**Summary:**

This paper proposes cSSDA integrates von Mises-Fisher (vMF) into the contrastive learning framework to address the semi-supervised domain adaptation (SSDA).

**Strengths:**

The paper is well-written, making the content easy to understand.

**Weaknesses:**

Based on my evaluation, I think that the paper's contribution is limited, as it suffers from insufficient novelty, an unconvincing methodology, a lack of comparison with recent approaches, weak performance, and insufficient experiment.

1.	Insufficient novelty: The application of von Mises-Fisher contrast has become quite common in domain adaptation research [1, 2, 3], and this work merely extends it to the semi-supervised domain adaptation setting. Moreover, I observe that prior work [6] has already explored integrating SCL [5] with the von Mises-Fisher contrast.

2.	Unconvincing methodology: As discussed in Section 2.6, this paper is an extension of the work in [7]; hence, it should emphasize the distinctions between this study and [7] to make the proposed method more convincing, rather than spending substantial space reiterating the theoretical analyses already presented in [5, 6].

3.	A lack of comparison with recent approaches: I noticed that the main evaluation table only compares with studies published before 2023, lacking comparisons with the more recent works from 2024 and 2025.

4.	Weak performance: I observed that the method performs poorly under the same experimental settings, failing to surpass the latest approaches [8, 9] and even underperforming compared to works [10, 11] published before 2023.

5.	Insufficient experiment: In previous SSDA studies [7-11], Office, Office-Home, DomainNet, and VisDA-2017 are commonly used datasets. Most works conduct comparisons on three or more datasets across different backbone networks, whereas this paper only performs a simple evaluation using ResNet-34 on Office-Home and DomainNet, which is clearly insufficient.

[1] vMFNet: Compositionality Meets Domain-generalised Segmentation. In MICCAI 2022.
[2] SimProF: A Simple Probabilistic Framework for Unsupervised Domain Adaptation. In AAAI 2025.
[3] Cross-modality domain adaptation for mechanical anomaly detection: A von mises-fisher VAE with enhanced interpretability. In Expert Syst. Appl. 2025.
[5] Supervised contrastive learning. In NeurIPS 2020.
[6] Probabilistic Contrastive Learning for Long-Tailed Visual Recognition. In TPAMI 2024.
[7] Semi-supervised domain adaptation with source label adaptation. In CVPR 2023.
[8] Enhancing Semi-supervised Domain Adaptation via Effective Target Labeling. In AAAI 2024.
[9] Inter-domain mixup for semi-supervised domain adaptation. In PR 2024.
[10] Semi-supervised Domain Adaptation via Prototype-based Multi-level Learning. In IJCAI 2023.
[11] Multi-level Consistency Learning for Semi-supervised Domain Adaptation. In IJCAI 2022.

**Questions:**

Please refer to the weaknesses outlined above, the authors need to provide detailed clarifications for these issues.

---

### Official Review · Reviewer_Gt8R · 2025-11-01

**Soundness:** 2
**Presentation:** 3
**Contribution:** 2
**Rating:** 2
**Confidence:** 3

**Summary:**

This paper addresses the problem in contrastive learning for SSDA. They identify two issues: the balanced sampling requirement in the contrastive learning batch and the limitation to assessing labeled samples in the target domain. To address these problems, they leverage the idea from the von Mises-Fisher (vMF) distribution to find a better direction in embedding space for optimization.

**Strengths:**

The paper is easy to read, and I can easily assess the main contribution and the main idea of every part.\
The paper is theoretically sound, but more evaluation and comparison are needed.

**Weaknesses:**

This paper's citation is out of date, so this submission cannot be further considered. Please add more recent works proposed in 2024 and 2025 for analysis and comparison, especially in the experiment part. For example, the belowed missing reference:
1. Ngo, Ba Hung, et al. "Learning cnn on vit: A hybrid model to explicitly class-specific boundaries for domain adaptation." Proceedings of the IEEE/CVF Conference on Computer Vision and Pattern Recognition. 2024.
2. Basak, Hritam, and Zhaozheng Yin. "Forget More to Learn More: Domain-Specific Feature Unlearning for Semi-supervised and Unsupervised Domain Adaptation." European Conference on Computer Vision. Cham: Springer Nature Switzerland, 2024.
3. Basak, Hritam, and Zhaozheng Yin. "SemiDAViL: Semi-supervised Domain Adaptation with Vision-Language Guidance for Semantic Segmentation." Proceedings of the Computer Vision and Pattern Recognition Conference. 2025.
4. Wu, Heng, et al. "Dara: distribution-aware representation alignment for semi-supervised domain adaptation in image classification." The Journal of Supercomputing 81.2 (2025): 376.

**Questions:**

This paper's citation is out of date, so this submission cannot be further considered. Please add more recent works proposed in 2024 and 2025 for analysis and comparison, especially in the experiment part. For example, the belowed missing reference:
1. Ngo, Ba Hung, et al. "Learning cnn on vit: A hybrid model to explicitly class-specific boundaries for domain adaptation." Proceedings of the IEEE/CVF Conference on Computer Vision and Pattern Recognition. 2024.
2. Basak, Hritam, and Zhaozheng Yin. "Forget More to Learn More: Domain-Specific Feature Unlearning for Semi-supervised and Unsupervised Domain Adaptation." European Conference on Computer Vision. Cham: Springer Nature Switzerland, 2024.
3. Basak, Hritam, and Zhaozheng Yin. "SemiDAViL: Semi-supervised Domain Adaptation with Vision-Language Guidance for Semantic Segmentation." Proceedings of the Computer Vision and Pattern Recognition Conference. 2025.
4. Wu, Heng, et al. "Dara: distribution-aware representation alignment for semi-supervised domain adaptation in image classification." The Journal of Supercomputing 81.2 (2025): 376.

---

### Official Review · Reviewer_EuXf · 2025-11-03

**Soundness:** 2
**Presentation:** 2
**Contribution:** 2
**Rating:** 2
**Confidence:** 4

**Summary:**

The paper focuses on the semi-supervised domain adaptation problem. While prior works often struggle with class imbalance, this work addresses the issue by proposing a novel contrastive learning framework that leverages the von Mises–Fisher (vMF) distribution to formulate a new contrastive loss to mitigate the imbalance challenge in domain adaptation.

**Strengths:**

1. It is interesting to investigate semi-supervised domain adaptation from the perspective of contrastive learning.
2. The method can yield promising results.

**Weaknesses:**

1. In lines 95–96, the authors state that their method is motivated by an observation, but no supporting reference or evidence is provided to justify this claim.

2. The proposed method is not clearly illustrated. In the methodology section, the unit-norm feature
$z$ is used multiple times, but it is unclear whether these features are generated from the labeled source data or the target domain. This lack of clarification makes the method ambiguous and difficult for readers to follow.

3. Some notations are not well explained. For instance, in Equations (7)-(11), the meanings of several symbols such as $\hat{\mu}$, $\hat{\lambda}$ and $\hat{\kappa}$ are unclear.

4. The online updating scheme for $\bar{z}$ is not clear.

5. The method is evaluated on only two benchmarks, which is insufficient to fully validate its effectiveness.

**Questions:**

see weakness.

---

### Meta-Review · Area_Chair_dgEb · 2025-12-27

**Summary:**

The paper proposes a cSSDA algorithm based on the von Mises–Fisher distribution to address class imbalance in SSDA. However, all three reviewers note that the method is not compared with recent SSDA methods and does not report results on additional standard benchmarks (e.g., VisDA-2017). Moreover, they also find the reported performance unconvincing, and one reviewer points out that using the vMF distribution is not novel, as it has been explored previously in UDA. Taken together, these concerns lead all reviewers to recommend rejection.

**Reviewer Concerns:**

No rebuttal was submitted, so the reviewers' concerns remain unaddressed.

**Reviewer Scores:**

Since the authors did not submit a rebuttal, it is very unlikely that any of the reviewers will change their score.

---

### Decision · Program_Chairs · 2026-01-26

Reject